# A Cross-Sectional Comparative Study: Could Asprosin and Peptide Tyrosine-Tyrosine Be Used in Schizophrenia to Define the Disease and Determine Its Phases?

**DOI:** 10.3390/diagnostics15050632

**Published:** 2025-03-05

**Authors:** Elif Özcan Tozoğlu, Nilifer Gürbüzer, Alev Lazoğlu Özkaya, Sümeyya Akyıldırım

**Affiliations:** 1Department of Psychiatry, Erzurum Faculty of Medicine, University of Health Sciences, Erzurum 25240, Türkiye; nilifer.gurbuzer@sbu.edu.tr; 2Department of Biochemistry, Erzurum City Hospital, Erzurum 25240, Türkiye; alevozkaya3113@gmail.com; 3Department of Psychiatry, Elazığ Mental Health and Diseases Hospital, Elazig 23100, Türkiye; dr.sumeyyaakyildirim@hot-mail.com

**Keywords:** asprosin, peptide tyrosine–tyrosine, schizophrenia, energy metabolism, marker

## Abstract

**Background/Objectives**: We aimed to evaluate asprosin and peptide tyrosine–tyrosine (PYY) levels in schizophrenia patients and the relationships between these levels and clinical severity, as well as whether these two hormones have a role in determining the disease and/or the phases of the disease. **Methods**: This study included 50 patients with schizophrenia in the remission phase, 50 in the acute phase, and 50 controls. The Positive and Negative Syndrome Scale (PANSS) was filled out for patients. The patients’ biochemical parameters and asprosin and PYY levels were measured. **Results**: Levels of asprosin and PYY were significantly different in all three groups (*p* < 0.001, *p* < 0.001). In the remission phase group, asprosin levels had a negative effect on PANSS general symptomatology scores (*p*: 0.002, *p* < 0.001). In the acute phase group, while PYY levels showed a negative effect on PANSS general symptomatology scores (*p*: 0.031), asprosin levels had a negative effect on all subscales of PANSS (*p* < 0.001). In the acute phase, a one-unit decrease in asprosin levels was associated with a 93% increase in PANSS total scores. The results of the receiver operating characteristic (ROC) analysis to distinguish the acute phase showed that PYY could not be used for diagnosis (*p*: 0.066), but asprosin was associated with the acute phase of schizophrenia (*p* < 0.001) and both asprosin and PYY were associated with the disease (*p* < 0.001, *p* < 0.001). **Conclusions**: We think that both asprosin and PYY can be used as potential biomarkers to identify schizophrenia, and only asprosin to identify the phases of the disease. PYY and asprosin levels may be markers that can be used to determine clinical severity.

## 1. Introduction

Schizophrenia is a complex psychiatric disorder and the underlying molecular mechanisms are still not fully understood. In addition to the classical dopaminergic and glutaminergic hypotheses, recent studies show mitochondrial dysfunction in the pathophysiology of schizophrenia, leading to disruption of energy homeostasis [1]. There is even experimental evidence showing that this mitochondrial dysfunction is associated with dopamine activation in the brain [2]. Probably as a clinical reflection of this situation, non-homeoestatic, hedonic eating behavior is observed in patients. The fact that irregular eating behaviors indicating disruption in energy homeostasis were reported before the advent of antipsychotics and in antipsychotic-naïve individuals essentially lends weight to the conclusion that the psychopathology of the disease may play a role in this situation. It has been suggested that schizophrenia may be associated with mitochondrial dysfunction in both the central nervous system and peripheral tissues and therefore, general deterioration in energy metabolism [3,4,5].

The nucleus arcuatus in the hypothalamus plays an important role in maintaining energy homeostasis. There are two distinct neuronal systems in the nucleus arcuatus that work in opposition to each other. Activation of neuropeptide Y and agouti-related peptide (AgRP) increases appetite. Pro-opiomelanocortin (POMC), a transcript peptide, decreases appetite [6].

Anorexigenic (obestatin, PYY, PP, GLP1, leptin) and orexigenic (ghrelin, AgRP, NPY, orexin, asprosin) peptides and hormones secreted from the gastrointestinal tract, adipose tissue, and central nervous system play a critical role in maintaining energy balance through various pathways. The asprosin in question is orexigenic protein hormone, which is produced and released by adipocytes in white adipose tissue during fasting. Asprosin crosses the blood–brain barrier and then increases the activity of AgRP neurons located in the arcuate nucleus. This signal inhibits the activity of POMC neurons in a γ-aminobutyric acid (GABA)-dependent manner [7]. Thus, nutrient intake is stimulated and energy homeostasis is regulated [8]. It has been reported that patients with lipodystrophy with asprosin deficiency due to a mutation in the fibrillin-1 gene consume less food and are extremely weak [8]. A study conducted with bulimia nervosa patients found that asprosin levels were significantly higher than in healthy controls, and it was shown that binge eating and loss of eating control increased with an increase in asprosin concentration [9]. In a study conducted on attention deficit hyperactivity disorder, which is a neurodevelopmental disorder, asprosin and AgRP levels were found to be significantly lower compared with controls [10]. This finding was interpreted as an indicator of disrupted homeostasis and could also result from a structural disorder. AgRP levels have been studied in patients with bipolar disorder, and many structural and genetic similarities with schizophrenia have been identified [11]. AgRP levels have been found to be low in patients with manic episodes, which has been interpreted as an indicator of impaired energy hemostasis [12].

Peptide tyrosine–tyrosine (PYY) is a gut hormone released from cells in the ileum and colon in response to feeding. Physiologically, PYY is associated with the “termination” signal for eating. Its anorexigenic effect primarily occurs by inhibiting orexigenic effects in the arcuate nucleus and activating pro-opiomelanocortin neurons. In patients with irritable bowel syndrome, PYY concentration and PYY cell count are reduced in the colon, indicating abnormalities in PYY related to irritable bowel syndrome pathology [13]. In a study of antipsychotic-naïve schizophrenia patients, PYY levels in their cerebrospinal fluid were found to be lower compared with controls [14]. This situation has been interpreted as suggesting that PYY may play a role in the pathogenesis of schizophrenia and could potentially serve as a trait marker.

In schizophrenia, the deterioration of this energy metabolism may be different during remission and acute periods. It has been thought that the presence of markers indicating this metabolism may be an easy and useful way to distinguish the stages of the disease, as well as providing a way to determine the clinical severity according to the levels of these markers and to monitor improvement during follow-up treatment.

The pathophysiological foundations of schizophrenia are not fully understood, and deepening the understanding of the biological mechanisms underlying this complex disease is an imperative need. As far as we know, no study has not yet been conducted in the literature to investigate the relationship between plasma asprosin and PYY levels, metabolic parameters, and psychopathology in patients with schizophrenia. Evaluation of the relationship between the severity and phases of schizophrenia, serum asprosin and PYY levels, and metabolic parameters can contribute to our knowledge about hedonic/homeostatic mechanisms in schizophrenia and help us understand the neurobiology of the disease.

In our study, it was planned to compare the levels of asprosin, PYY, body mass index (BMI), and metabolic parameters such as lipid profiles, blood sugar, glycated hemoglobin (HbA1c) between schizophrenia patients in the acute phase, schizophrenia patients in remission, and healthy controls without additional psychopathology or metabolic disease. We aimed to investigate whether asprosin and PYY could be biomarkers that could identify the disease or phases of the disease, as well as whether the data obtained were related to clinical appearance using PANNS scores, a scale that measures the positive symptoms, negative symptoms, and general psychopathology levels in schizophrenia patients.

## 2. Materials and Methods

### 2.1. Procedures

Our study focused on the cross-sectional clinical characteristics of the participants, routine biochemical parameters, and analysis of serum fasting PYY and asprosin. The diagnosis of schizophrenia was made using the *Diagnostic and Statistical Manual of Mental Disorders 5* (DSM-5) clinician version (SCID-5/CV). Additional psychopathologies in psychiatric patients and psychopathology in the control group were excluded by SCID-5/CV. The Positive and Negative Symptoms Scale (PANSS) was filled in by psychiatrists for the patients. The clinical and sociodemographic data of the participants were recorded. Fasting blood samples were taken from all participants between 08:00 and 10:00. The weight and height of each participant was measured and BMI was calculated.

In order to avoid bias, participants were evaluated twice by two different psychiatrists at different times and strict inclusion/exclusion criteria were applied.

### 2.2. Ethics and Consent

The research protocol was approved by the Scientific Research Ethics Committee of Health Sciences University Erzurum Faculty of Medicine (Erzurum, Türkiye) with the decision numbered BAEK 2023/07-82 and carried out in accordance with the Helsinki Declaration. The registration number of the ethical approval is 22943. Written informed consent was obtained from all participants and/or their court-appointed guardians, if any, and from the guardians of patients in the acute phase, stating that they agreed to participate in the study and gave permission to publish all clinical and other data in the article.

### 2.3. Research Sample

In the studies on asprosin and PYY in attention deficit hyperactivity disorder and schizophrenia patients, similar studies in the literature were taken as a basis and the G*Power statistical program was used for sample calculation [10,15]. According to the sample calculations, it was determined that for a difference of 10 units to be significant, it would be appropriate to include at least 47 participants from each group at 95% power and 95% confidence interval (effect size = 0.70). Considering the possible loss of participants, it was planned to include 50 participants from each group. It was planned to include patients diagnosed with schizophrenia in the remission period or acute period who applied to Erzurum City Hospital Psychiatry Clinic and Polyclinic between November 2023 and April 2024. The control group was planned to consist of the same number of healthy volunteers.

The inclusion criteria for all participants were being between 18–65 years of age, have a BMI below 30, not having a diagnosed physical acute or chronic illness, not having an active alcohol or substance use disorder, not being pregnant or postpartum, agreeing to participate in the study and/or if they had a guardian, their guardian agreed to participate in the study, and the agreement of the guardians of patients in the acute phase for them to participate in the study. The inclusion criterion for schizophrenia patients in the remission phase was that they were in remission according to the Van Os criteria [16]. The presence of another additional psychopathology was accepted as an exclusion criterion for the patient group.

For this study, the possible participants included 92 patients with schizophrenia in the acute phase, 151 patients with schizophrenia in remission at the specified dates, and 83 healthy controls. Flow diagrams illustrating recruitment procedures are presented in Figure 1.

### 2.4. Data Collection Tools

#### 2.4.1. Sociodemographic and Clinical Data Form

We used a form developed by researchers to record the characteristics of participants, such as age, gender, duration of education, marital status, work status, alcohol and cigarette use, height, weight, body mass index, duration of illness, number of attacks.

#### 2.4.2. SCID-5-CV

The Structured Clinical Interview for DSM (SCID) is one of the most widely used diagnostic tools in clinical research worldwide. The latest version is SCID-5. SCID-5/CV is a comprehensive, standardized tool for the evaluation of major psychiatric disorders according to the definitions and criteria stated in DSM-5. The structured clinical interviews included 32 diagnostic categories with detailed diagnostic criteria and 17 diagnostic categories with research questions. The validity and reliability study of the Turkish version of SCID-5/CV has been confirmed [17].

#### 2.4.3. Positive and Negative Syndrome Scale (PANSS)

PANSS is a 30-item scale used by clinicians to assess symptom severity in psychotic disorders. It covers symptoms such as delusions, hallucinations, conceptual disorganization, mannerisms, blunted affect, social withdrawal, and lack of spontaneity. The scale consists of three subscales, each of which is scored independently. The score that can be obtained from the positive syndrome subscale varies between 7 and 49. The score that can be obtained from the negative syndrome subscale varies between 7 and 49. The score that can be obtained from the general psychopathology subscale varies between 16 and 112. Turkish validity and reliability were determined by Kostakoglu et al. and internal consistency was calculated to be between 0.71–0.75 [18].

### 2.5. Biochemical Analysis

Fasting blood samples were collected from the participants between 08:00 and 10:00 h, in a sitting position, after they had been allowed to rest. Samples were collected from the antecubital region by experienced personnel using a vacutainer. To obtain serum, a blood sample taken in a gel tube (Vacusera^®^) was coagulated for 30 min at room temperature and then centrifuged at 3000 rpm for 10 min. Alanine aminotransferase (ALT), aspartate aminotransferase (AST), glucose, cholesterol, highdensity lipoprotein (HDL), low-density lipoprotein (LDL), and triglyceride (TG) levels were determined spectrophotometrically using a Siemens Atelica clinical chemistry analyzer. For the measurement of HbA1c levels, blood samples were collected in hemogram tubes (Vacusera^®^, Disera Medical Supplies INC., Izmir, Türkiye) containing anticoagulant and measured using high-performance liquid chromatography in a Lifotronik H9 device. After routine analysis, serum samples were aliquoted and frozen at −80 °C until analysis. PYY and asprosin assays were performed in a single session at Erzurum City Hospital Medical Biochemistry Laboratory, using a RelAssay automated ELISA reader (BiobaseBiodusty, Shandong, Co., Ltd., Jinan, China) and commercially purchased BT Lab ELISA kits (Jiaxing, China) according to the standard protocol recommended by the manufacturer. The ELISA kits used were for research use only and the measurement ranges were 3–900 pg/mL for PYY and 0.5–100 ng/mL for asprosin. Several quality control methods were implemented to improve the accuracy and reliability of the biochemical analyses. All data and samples were meticulously tracked using sample codes during laboratory processes. Furthermore, the performance of the instruments was checked using quality control samples before each analysis session. Quality assurance was ensured by regularly calibrating the instruments and using standard protocols. The accuracy and reliability of the ELISA test kits were ensured by high-sensitivity testing performed in accordance with the protocol provided by the manufacturer. Measurements were repeated and consistent results were obtained, demonstrating the reliability of the data. There were no instances of missing data or poor-quality data.

### 2.6. Statistical Analysis

Analyses were conducted using IBM Statistical Package for Social Sciences (SPSS) version 26 (IBM Corp., Armonk, NY, USA) analysis software. The data were presented in terms of mean, standard deviation, and count. Normality analysis was performed to check whether the skewness and kurtosis values of all variables fell within the range of −2 to +2. These values indicated that the normality assumption was met [19]. Since normal distribution conditions were met, an independent samples test was used for comparison between two independent groups and one-way ANOVA testing was applied for comparisons between three independent groups. For post hoc analysis of significance between groups, Bonferroni correction was applied when the data were distributed homogeneously and Tamhane’s T2 correction was applied if homogeneity was not achieved. Chi square testing was applied for comparisons between categorical variables. Pearson correlation analysis was applied to correlate biochemical data with sociodemographic data and clinical scales. Linear regression analysis was applied to examine the effects of asprosin and PYY on the clinical appearance in both the acute-phase and remission-phase schizophrenia groups. Receiver operating characteristic curve (ROC) analysis was performed to determine whether the continuous variable could be used in diagnosis and phase determination and to determine cutoff values. The level of statistical significance was taken as *p* < 0.05.

## 3. Results

The study was completed with 50 patients with schizophrenia in remission, 50 patients with schizophrenia in the acute phase, and 50 controls of similar age and gender. Skewness and kurtosis were assessed to determine the normality of the continuous variables. In Table 1, the participants’ sociodemographic variables and the laboratory results are shown comparatively for the acute-phase schizophrenia patients, remission-phase schizophrenia patients, and control groups. All of the patients in the acute phase included in this study had discontinued their medication for an unknown period of time and were not currently taking any medication. The medications used by patients in the remission phase are shown in Table 1.

In the one-way ANOVA test conducted for all three groups, post hoc analysis was performed on the data where significance was detected. The pairwise comparisons and post hoc analysis results for all three groups are presented in Table 2.

Although the lipid profiles were similar across all three groups, the levels of asprosin and PYY were significantly different in each group (*p* < 0.001, *p* < 0.001). HbA1c values were statistically higher only in the acute phase group compared with the remission phase and the control group, while fasting blood glucose levels were significantly elevated in both the acute-phase and remission-phase groups, both internally and compared with the control group.

The ALT value was significantly lower in the acute-phase group compared with the control group, whereas the AST value was significantly higher in the acute-phase group compared with both the remission and control groups. The AST/ALT ratio differed significantly among all three groups, with an average ratio exceeding 1 in the acute-phase group. In the remission group, BMI was statistically higher than both the acute-phase group and the control group, but no significant difference was observed between the control group and the acute-phase group.

Correlation analysis was performed separately in the remission and acute-phase groups in order to evaluate the relationship between asprosin and PYY values and clinical appearance (Table 3). Asprosin levels in the remission and acute-phase group showed a negative correlation with positive, negative, and general symptomatology PANSS scores. PYY values were negatively correlated with negative and general symptomatology PANSS scores in the remission and acute-phase group and with positive PANSS scores only during the acute phase.

To investigate the relationship between PANSS scores indicating clinical appearance and the levels of asprosin and PYY, linear regression analyses were performed separately for the remission-phase group and acute-phase group using PANSS scores as the dependent variable. In the remission-phase group, no significant effect of PYY levels on PANSS scores was observed, while asprosin levels showed a significant negative effect on both PANSS total and general symptomatology scores. In the acute-phase group, PYY levels were significantly and negatively associated with only PANSS general symptomatology scores, whereas asprosin levels had significant negative effects on both PANSS total scores and all the subscales. In the acute phase group, a decrease of one unit in asprosin levels was associated with a 93% increase in total PANSS scores, a 68% increase in positive scores, a 92% increase in negative scores, and a 97% increase in general symptomatology scores. The regression model data are presented in Table 4.

The results of the analysis to differentiate the acute phase show that PYY cannot be used to define the acute phase, but asprosin is associated with the acute phase in schizophrenia (Figure 2). Additionally, values below 7.39 for asprosin were found to be effective for defining the acute phase of schizophrenia.

The results of the analysis to differentiate disease showed that both PYY and asprosin were associated with disease (Figure 2). Additionally, according to the analysis results, values below 11.91 for asprosin and below 161.59 for PYY were effective for identifying the disease.

In patients with schizophrenia, those whose asprosin values were below or above 7.39 were divided into two groups (Groups 1 and 2, respectively). Clinical and laboratory data that were previously found to be significant were compared between the groups (Table 5). BMI was similar between groups, but PANSS scores were significantly higher in Group 1 than in Group 2. ALT and HbA1c values were similar between groups; AST and FBG values were found to be significantly higher in Group 1.

## 4. Discussion

One of the underlying pathophysiological mechanisms of schizophrenia is disruption in energy metabolism homeostasis. This condition may arise from central factors such as abnormalities in the hypothalamic–pituitary–adrenal (HPA) axis, as well as from disturbances in hormonal signals originating from peripheral tissues. Although we saw hedonic-style eating patterns in our patients that did not meet the clinical appearance of an eating disorder criteria, interestingly, we found in the results of our study that levels of asprosin, a hormone known to be an appetite stimulant released from adipose tissue, were significantly lower in schizophrenia patients compared with healthy controls, and this decrease was more pronounced in acute-phase patients. However, in a study conducted with bulimia nervosa patients, asprosin levels were found to be higher than in healthy controls [9]. In our study, the known anorexegenic PYY secreted in the intestine was found to be lower in patients than in controls, and this decrease was more pronounced in patients in the acute phase. In schizophrenia, which is a developmental disorder, this impairment in energy metabolism probably occurs both in the axes in the brain and in the associated peripheral tissues. Although the two hormones work inversely, the positive correlation of both in patients may indicate a malfunction in this axis.

In a study in the research literature, the PYY levels in the cerebrospinal fluid of 35 schizophrenia patients were found to be low compared with healthy controls, which was interpreted as a marker for PYY disease [14]. In another study conducted with schizophrenia patients taking clozapine, PYY levels were again found to be low, which was more associated with BMI [15]. However, in our study, we evaluated schizophrenia patients both in the remission phase and in the acute phase. Although the BMI of patients in the acute phase was similar to healthy controls, we found that PYY levels were significantly lower.

In a study conducted with patients with attention deficit hyperactivity disorder, which is also a neurodevelopmental disorder, asprosin levels were found to be significantly lower than in controls, and this was interpreted as an indicator of impaired homeostasis and also as a result of a structural disorder [10]. Compared with controls, asprosin-related AgRP levels were also found to be lower in patients with bipolar disorder, which has many structural and clinical aspects in common with schizophrenia, and this was interpreted as an indicator of impaired homeostasis [12].

In our study, low asprosin and PYY levels were significantly correlated with PANSS scores, which indicate the severity of clinical presentation. Disruption in the system is likely reflected in the clinic, especially in negative symptoms and general symptomatology. Gottschalk and colleagues showed that schizophrenia is associated with a hypoglutamatergic state and hypofunction of energy metabolism [20]. In other words, deterioration in energy metabolism and exacerbation of clinical symptomatology can be evaluated in this context. In addition, in recent studies examining gastrointestinal hormones, including PYY, it was reported that these hormones have receptors expressed in brain regions that regulate not only hunger and energy metabolism but also stress, behavior, and cognitive function [21]. It has been found that PYY levels are positively related to gray matter volume but negatively related to postprandial activity in the caudate nuclei, and caudate activity is negatively related to cerebral blood flow in the prefrontal and paralimbic regions involved in reward behavior [22]. In brain imaging studies conducted in patients with schizophrenia, decreases in gray matter and activation of the limbic and paralimbic regions are frequently reported to be associated with hallucinations in schizophrenia. In light of this information, the significant decrease in PYY levels in our study suggests that this may be the result of a structural disorder.

According to the neurodevelopmental hypothesis, brain damage occurs in the early stages of neuronal development due to a combination of genetic and early developmental factors. For this reason, the normal maturation process of the brain is disrupted and neuron development is negatively affected. Damage may occur during the stages of cell proliferation, differentiation, cell migration, synaptic pruning, or programmed cell formation. It is thought that the disruptions in signal transmission and neuron circuits that may occur as a result of all these effects within the development of the nervous system can lead to the emergence of schizophrenia symptoms. Presumably, these disruptions also lead to disturbances in energy metabolism; the resulting low levels of asprosin and PYY in patients correlated with the severity of their clinical condition and were detected at levels lower than those in healthy controls. In our study, asprosin levels were found to be associated with the disease and were also found to be different enough to be an indicator of the acute phase. However, the data obtained were not at a level that can be presented as direct evidence.

One of the processes disrupted in schizophrenia is glucose metabolism. Lower rates of glucose metabolism have been detected in the hippocampus and anterior cingulate cortex of schizophrenia patients compared with healthy controls [23]. Numerous studies conducted with schizophrenia patients have shown that hyperglycemia, impaired glucose tolerance, and/or insulin resistance occur even in first-episode, previously untreated, antipsychotic-naïve schizophrenia patients [24]. In our study, although the patients were not previously diagnosed with diabetes, FBG levels were found to be highest in patients in the acute phase and statistically significantly higher in patients in the remission phase compared with controls. HbA1c values, which provide information about three-month blood sugar levels, were also significantly higher in the acute-phase patient group. The results of our study also indicated the impairment of glucose metabolism in schizophrenia, which is thought to be the result of a neurodevelopmental disorder.

In our study, AST and ALT values were among the reference values. Although AST was not statistically significant in patients in the remission phase but was higher than in the controls, it was statistically significantly higher in patients in the acute phase than those in remission or the controls. ALT values tended to decrease from the control to the acute phase, although they were still within the reference ranges. When evaluating the AST/ALT ratio, it was observed that there were statistically significant differences in all three groups, and values greater than 1 were detected in acute-phase patients. ALT is a cytosolic enzyme, whereas AST is found in both the cytosol and mitochondria. In schizophrenia, mitochondrial dysfunction has been implicated as a cause of energy metabolism impairment [25]. Specifically, damage to mitochondria in hepatocytes may lead to the release of mitochondrial AST, contributing to this condition. On the other hand, although not measured, vitamin B6 deficiency due to inappropriate eating habits in schizophrenia [26], especially in the acute phase, may possibly have led to a significant decrease in ALT levels compared with the controls. Considering that in schizophrenia, cognitive impairment persists even during remission despite being higher during the acute phase, the increase in the AST/ALT ratio is consistent with other studies [27].

In our study, BMI was statistically significantly higher only in patients in the remission phase compared with patients in the acute phase and the control group. Weight gain in schizophrenia has been found to increase with increasing age, both naturally and in relation to the disease. Among schizophrenia patients, the obesity rate was found to be 10.9% in those under the age of thirty, 15.6% in the 30–39 age range, 20.9% in the 40–49 age range, and 22.9% in the 50–59 age range [28]. Although the average age of each of the three groups was statistically similar in our study, the average age was higher in the remission group compared with the other two groups. It was thought that the high BMI in the remission group was related to the average age.

Considering that metabolic disorders, and therefore, abnormal lipid profiles, are common in patients with schizophrenia, it is an interesting finding that their lipid profile was similar to healthy controls in our study. Already, various studies conducted on the lipid profiles of patients with schizophrenia have emphasized the complexity and importance of this issue. Many studies have shown that people with schizophrenia usually have higher serum lipid levels (cholesterol and TG) than the healthy population [29,30]. This is usually considered to be a result of the use of antipsychotic drugs and lifestyle factors. However, dyslipidemia has also been shown in antipsychotic-naïve schizophrenia patients. In a previous study, it was shown that serum triglyceride levels were significantly higher in patients with schizophrenia than in healthy controls, but serum cholesterol levels were similar, in parallel with our study [31]. In addition, again in that study, the lipid levels of patients remained stable over time [31]. These findings suggest that lipid profiles may be related to the pathophysiology of schizophrenia. The lipid profiles of patients can be influenced by a number of factors, such as the nature of the disease, treatment regimens, and individual biological differences. The difference in the results of our study may also be due to these reasons.

Various limitations should be taken into account when interpreting the results of this study. The cross-sectional design of the study was its main limitation. The cross-sectional design limited the ability to infer causality between asprosin and PYY levels and disease. The cross-sectional design did not provide data on the temporal dynamics of these biomarkers. Conducting a study in patients with first-attack schizophrenia in order to observe the disease-defining effect of asprosin and PYY will give more reliable results. Again, designing a study in which values from the same patient in both the acute phase and the remission phase can be compared can provide a more accurate interpretation of the data. In a study examining PYY in patients with schizophrenia, patients who received neuroleptics and those who did not were evaluated; it was stated that using neuroleptics had no effect on PYY values [14]. Although we argue that the deterioration in asprosin and PYY regulation in our study was due to structural reasons, the fact that patients had been taking antipsychotics may still have been a confusing factor. Although patients in the acute phase had discontinued the drug at the time of recruitment, it remains unclear how long they had been off the drug. Conducting this study in drug-free patients or designing the study in such a way that drug-related factors can be controlled may provide clearer interpretations. Supporting these findings, which we think may be the result of a structural disorder, with neuroimaging studies may place the current study on stronger foundations. In our study, no evaluation was made regarding the patients’ nutritional styles. When evaluating appetite-related hormones, studies conducted with groups subjected to standard diets may provide more accurate data. Physical activity is known to affect metabolic parameters [32]. However, the physical activity levels of the patients were not evaluated in our study. This may be a confounding factor affecting the results. Participants with known physical disease and participants with a BMI above 30 were excluded from our study. However, possible unknown, undiagnosed metabolic diseases may have been missed during routine examinations. Although the anorectic effects of PYY are largely attributed to PYY3-36, we measured total PYY concentrations in our study. The two subtypes, PYY1–36 and PYY3-36, have different receptor selectivity profiles and may have influenced the study results [21]. Although obesity and metabolic dysfunction are common in schizophrenia, the reasons that both asprosin, an orexigenic peptide, and PYY, an anorexigenic peptide, are low should be addressed as a separate research topic. Studies on the subject at the receptor level may be useful. On the other hand, the Turkish validity and reliability of the scales used, evaluation of the participants by two different psychiatrists, the fact that the study was conducted considering many variables together with asprosin and PYY, and the use of an appropriately selected control group are the strengths of the current research. Our study separately evaluated schizophrenia patients in remission and the acute phase and found that asprosin and PYY are disease markers, providing valuable evidence that asprosin may be a determinant of the phase of disease.

## 5. Conclusions

Schizophrenia is a multifaceted mental illness characterized by a variety of symptoms that affect cognition, perception, and behavior. Understanding the progression of the disease and identifying its stages are crucial for effective management and treatment. Research into various biomarkers and neuroanatomic markers to identify the disease of schizophrenia is constantly on the agenda. We think that asprosin, a new hormone associated with energy metabolism, may be a potential biomarker for identifying the phases of the disease and that both asprosin and PYY may be potential biomarkers for identifying schizophrenia disease. On the other hand, PYY and asprosin levels may be markers that can be used to determine the clinical severity of the disease. To confirm the findings of this study, more comprehensive independent cohort studies are needed in the future to elucidate the underlying mechanisms, especially by addressing possible confounding factors and assessing the limitations of the current study.

## Figures and Tables

**Figure 1 diagnostics-15-00632-f001:**
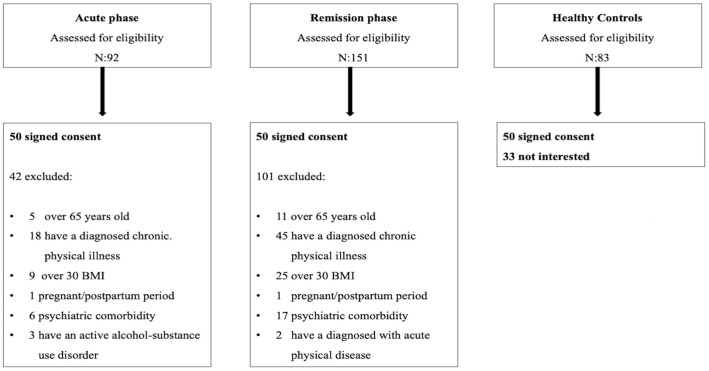
Flow diagram of procedure.

**Figure 2 diagnostics-15-00632-f002:**
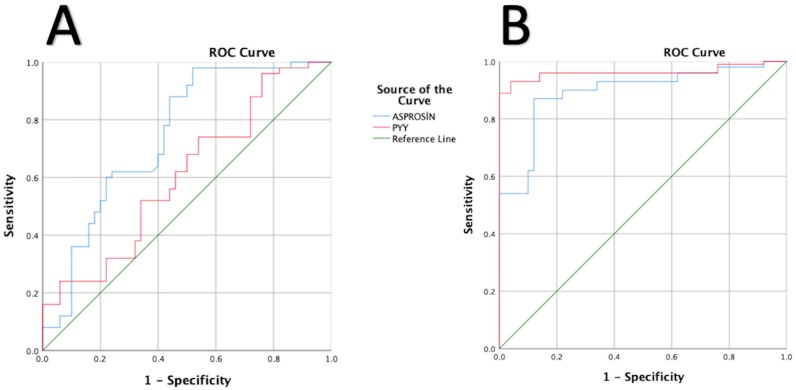
(**A**) ROC analysis to distinguish the acute phase, (**B**) ROC analysis to distinguish disease. Notes: (**A**) Asprosin, 95% confidence interval 0.647–0.843, area: 0.745, std. error: 0.50, *p* < 0.001, cutoff value: 7.39, sensitivity: 62%, specificity: 76%; PYY, 95% confidence interval 0.496–0.504, area: 0.607, std. error: 0.56, *p*: 0.066. (**B**) Asprosin, 95% confidence interval 0.840–0.946, area: 0.893, std. error: 0.27, *p* < 0.001, cutoff value: 11.91, sensitivity: 87%, specificity: 88%; PYY, 95% confidence interval 0.930–0.996, area: 0.962, std. error: 0.16, *p* < 0.001, cutoff value: 161.59, sensitivity: 93%, specificity: 96%.

**Table 1 diagnostics-15-00632-t001:** Comparison of sociodemographic, laboratory data, and scale scores of the groups.

		Control *N*: 50	Remission Phase *N*: 50	Acute Phase *N*: 50	*p* Value
Sex	women	28	19	23	0.195
	men	22	31	27	
Age(Year)		39.08 ± 11.63	43.30 ± 11.58	39.76 ± 10.35	0.133
Marital Status	single	12	21	28	*p* < 0.001
	married	37	15	19	
	divorced, widowed	1	14	3	
Education Time (Year)		11.12 ± 2.46	7.44 ± 3.68	7.08 ± 2.60	*p* < 0.001
Smoking	yes	29	39	36	0.084
	no	21	11	14	
Alcohol Use	yes	7	1	0	0.003
	no	43	49	50	
Substance Use History	yes	0	2	10	*p* < 0.001
	no	50	48	40	
BMI (kg/m^2^)		25.35 ± 3.36	27.13 ± 3.11	24.86 ± 3.00	*p* < 0.001
Asprosin (ng/mL)		28.26 ± 17.67	13.78 ± 12.34	7.37 ± 2.58	*p* < 0.001
PYY (pg/mL)		271.18 ± 104.02	116.70 ± 72.63	88.72 ± 25.68	*p* < 0.001
ALT (U/L)		25.68 ± 14.66	21.26 ± 10.47	18.88 ± 10.20	0.017
AST (U/L)		14.26 ± 4.25	16.70 ± 8.28	23.44 ± 8.73	*p* < 0.001
AST/ALT Ratio		0.64 ± 0.23	0.83 ± 0.40	1.42 ± 0.53	*p* < 0.001
TG (mg/dL)		154.64 ± 126.88	155.70 ± 108.96	137.28 ± 67.59	0.612
HDL (mg/dL)		41.68 ± 10.59	39.88 ± 10.48	42.06 ± 9.17	0.515
LDL (mg/dL)		126.44 ± 34.71	121.56 ± 38.84	121.82 ± 34.92	0.751
Cholesterol (mg/dL)		178.48 ± 34.73	164.02 ± 42.45	167.68 ± 33.98	0.134
FBG (mg/dL)		82.38 ± 11.69	95.96 ± 18.43	112.26 ± 16.82	*p* < 0.001
HbA1c (%)		5.14 ± 0.64	5.04 ± 0.64	5.41 ± 0.49	0.006
Duration of illness (Years)			16.44 ± 9.07	14.44 ± 8.53	0.259
Number of exacerbations			7.92 ± 5.75	9.42 ± 5.99	0.205
PANSS positive			11.78 ± 3.50	37.40 ± 5.75	*p* < 0.001
PANSS negative			14.02 ± 4.67	35.34 ± 8.98	*p* < 0.001
PANSS general psychopathology			31.66 ± 10.47	102 ± 5.14	*p* < 0.001
Drug use					
Paliperidone palmitate once monthly, LAI formulation			19	-	
Paliperidone palmitate 3-monthly, LAI formulation			7	-	
Aripiprazole once monthly, LAI formulation			13	-	
Olanzapine			8	-	
Risperidone			5	-	
Quetiapin			23	-	
Clozapine			9	-	
Amisulpride			5	-	

Data are expressed as mean ± SD or *N*. Chi square values are presented for categorical data and f value for numerical data. ALT: Alanine aminotransferase; AST: Aspartate aminotransferase; BMI: Body mass index; FBG: Fasting blood glucose; HbA1c: Hemoglobin A1c; HDL: High-density lipoprotein; LAI: Long-acting injectable; LDL: Low-density lipoprotein; PANSS: Positive and Negative Syndromes Scale; PYY: Peptide tyrosine–tyrosine.

**Table 2 diagnostics-15-00632-t002:** Post hoc Analysis Results.

	*p* Value		
	Control–Remission Phase	Control–Acute Phase	Acute Phase–Remission Phase
Asprosin (ng/mL)	*p* < 0.001	*p* < 0.001	0.002
PYY (pg/mL)	*p* < 0.001	*p* < 0.001	0.038
ALT (U/L)	0.237	0.025	0.582
AST (U/L)	0.19	*p* < 0.001	*p* < 0.001
AST/ALT Ratio	0.016	*p* < 0.001	*p* < 0.001
FBG (mg/dL)	*p* < 0.001	*p* < 0.001	*p* < 0.001
HbA1c (%)	1	0.069	0.006
BMI (kg/m^2^)	0.017	1	0.001

ALT: Alanine Aminotransferase; AST: Aspartate Aminotransferase; BMI: Body mass index; FBG: Fasting blood glucose; HbA1c: Hemoglobin A1c; PYY: Peptide tyrosine–tyrosine.

**Table 3 diagnostics-15-00632-t003:** Correlation analysis of asprosin and PYY levels according to disease phase, clinical severity, and laboratory data.

		Remission Phase		Acute Phase	
		Asprosin	PYY	Asprosin	PYY
Age (Year)	r	−0.164	−0.214	−0.119	0.290 *
	*p*	0.255	0.136	0.409	0.041
Duration of illness (years)	r	−0.369 **	−0.367 **	−0.448 **	−0.207
	*p*	0.008	0.009	0.001	0.149
Number of exacerbations	r	−0.254	−0.266	−0.476 **	−0.450 **
	*p*	0.075	0.062	0.000	0.001
PANSS positive	r	−0.318 *	−0.230	−0.663 **	−0.288 *
	*p*	0.025	0.107	0.000	0.042
PANSS negative	r	−0.670 **	−0.658 **	−0.857 **	−0.315 *
	*p*	0.000	0.000	0.000	0.026
PANSS general psychopathology	r	−0.700 **	−0.637 **	−0.890 **	−0.318 *
	*p*	0.000	0.000	0.000	0.025
ALT (U/L)	r	0.341 *	0.403 **	−0.060	0.314 *
	*p*	0.015	0.004	0.681	0.026
AST (U/L)	r	0.209	0.218	−0.212	−0.033
	*p*	0.146	0.127	0.139	0.818
TG (mg/dL)	r	0.093	0.118	−0.289 *	−0.023
	*p*	0.521	0.416	0.042	0.873
HDL (mg/dL)	r	−0.098	−0.069	0.299 *	0.215
	*p*	0.497	0.636	0.035	0.134
LDL (mg/dL)	r	0.228	0.148	0.155	0.455 **
	*p*	0.111	0.304	0.283	0.001
Cholesterol (mg/dL)	r	0.165	0.160	0.110	0.289 *
	*p*	0.252	0.268	0.445	0.042
FBG (mg/dL)	r	0.078	0.026	0.124	0.119
	*p*	0.590	0.859	0.390	0.412
HbA1c (%)	r	0.133	0.188	0.187	0.216
	*p*	0.357	0.191	0.194	0.132

r: Pearson’s correlation coefficient. **: Correlation is significant at the 0.01 level (2-tailed). *: Correlation is significant at the 0.05 level (2-tailed). ALT: Alanine aminotransferase; AST: Aspartate aminotransferase; FBG: Fasting blood glucose; HbA1c: Hemoglobin A1c; HDL: High-density lipoprotein; LDL: Low-density lipoprotein; PANSS: Positive and Negative Syndromes Scale; PYY: Peptide tyrosine–tyrosine; TG: Triglyceride.

**Table 4 diagnostics-15-00632-t004:** Determinant variables in the linear regression model in which the PANSS score was considered as the dependent variable.

								95% Confidence Interval		Model Summary	
Dependent Variables	Group	Variables	B	SE	BETA	t	*p*	Lower Bound	Upper Bound	R	Durbin-Watson
PANSS total score	Remission Phase	(Constant)	69.875	3.123		22.373	0.000	63.592	76.158	0.753	1.400
		Asprosin	−1.005	0.311	−0.806	−3.228	0.002	−1.632	−0.379		
		PYY	0.012	0.053	0.058	0.233	0.817	−0.094	0.119		
	Acute Phase	(Constant)	215.714	5.069		42.553	0.000	205.516	225.912	0.873	2.012
		Asprosin	−6.691	0.589	−0.930	−11.354	0.000	−7.877	−5.506		
		PYY	0.094	0.059	0.130	1.593	0.118	−0.025	0.214		
PANSS positive	Remission Phase	(Constant)	12.163	1.009		12.057	0.000	10.134	14.193	0.357	2.095
		Asprosin	−0.202	0.101	−0.711	−2.004	0.051	−0.404	0.001		
		PYY	0.021	0.017	0.426	1.201	0.236	−0.014	0.055		
	Acute Phase	(Constant)	47.679	2.399		19.871	0.000	42.852	52.506	0.665	1.898
		Asprosin	−1.535	0.279	−0.689	−5.502	0.000	−2.096	−0.973		
		PYY	0.012	0.028	0.052	0.418	0.678	−0.045	0.068		
PANSS negative	Remission Phase	(Constant)	18.235	1.061		17.179	0.000	16.100	20.370	0.678	1.362
		Asprosin	−0.161	0.106	−0.426	−1.525	0.134	−0.0374	0.051		
		PYY	−0.017	0.018	−0.265	−0.948	0.348	−0.053	0.019		
	Acute Phase	(Constant)	54.669	2.505		21.820	0.000	49.629	59.709	0.866	2.002
		Asprosin	−3.227	0.291	−0.928	−11.078	0.000	−3.813	−2.641		
		PYY	0.050	0.029	0.144	1.720	0.092	−0.009	0.109		
PANSS general psychopathology	Remission Phase	(Constant)	39.477	2.306		17.118	0.000	34.837	44.116	0.700	1.663
		Asprosin	−0.642	0.230	−0.757	−2.793	0.008	−1.105	−0.180		
		PYY	0.009	0.039	0.061	0.227	0.822	−0.070	0.087		
	Acute Phase	(Constant)	113.365	1.246		91.002	0.000	110.859	115.872	0.901	2.213
		Asprosin	−1.930	0.145	−0.970	−13.326	0.000	−2.221	−1.639		
		PYY	0.032	0.015	0.162	2.220	0.031	0.003	0.062		

PANSS: Positive and Negative Syndromes Scale; PYY: Peptide tyrosine–tyrosine.

**Table 5 diagnostics-15-00632-t005:** Comparison of patient groups determined according to cutoff value.

	Group 1 (*N*: 43)	Group 2 (*N*: 57)	*p* Value
	Mean ± SD		
PANSS positive	32.42 ± 13.20	18.68 ± 10.93	*p* < 0.001
PANSS negative	35.65 ± 10.78	16.40 ± 6.59	*p* < 0.001
PANSS general psychopathology	88.49 ± 27.52	50.49 ± 33.59	*p* < 0.001
BMI (kg/m^2^)	25.96 ± 3.55	26.02 ± 3.04	0.92
ALT (U/L)	19.91 ± 10.06	20.19 ± 10.66	0.892
AST (U/L)	23.30 ± 10.42	17.63 ± 7.17	0.002
AST/ALT ratio	1.26	1.03	0.035
FBG (mg/dL)	107.60 ± 19.58	101.47 ± 18.96	0.118
HbA1c (%)	5.30 ± 0.59	5.17 ± 0.60	0.295

ALT: Alanine aminotransferase; AST: Aspartate aminotransferase; BMI: Body mass index; FBG: Fasting blood glucose; HbA1c: Hemoglobin A1c; PANSS: Positive and Negative Syndromes Scale; SD: Standard deviation.

## Data Availability

The authors are committed to making the data available if requested by the Journal.

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
