# Peer review of "A Cross-Sectional Comparative Study: Could Asprosin and Peptide Tyrosine-Tyrosine Be Used in Schizophrenia to Define the Disease and Determine Its Phases?"

_diagnostics, 2025, doi:10.3390/diagnostics15050632_

Round 1

Reviewer 1 Report

Comments and Suggestions for Authors

The authors’ investigated the asprosin and peptide tyrosine-tyrosine (PYY) levels in schizophrenia patients and the relationship of these levels with clinical severity and whether these two hormones have a role in determining the disease and/or the phases of the disease, and their results suggest such involvement. Experiments are well designed and documented with appropriate biochemical and clinical methods, relevant controls and statistical analysis. The manuscript is concisely written and supported by appropriate references, the results are clearly presented in tables and figures, and the drawn conclusions are valid.

Comments and suggestions:

1.     I recommend correcting the visualization of ROC curves., ROC curves of indicators should be above the reference line according to generally accepted data representations. It is not difficult, only they need to change the data in the Value of State Variable line.

2.     The authors indicate in the limitations section that therapy was not monitored. It is known that antipsychotic drugs significantly affect the concentrations of hormones of eating behavior, and metabolic syndrome is the main side effect of second-generation atypical antipsychotics. Given the long duration of schizophrenia in both groups, I would recommend separately presenting what percentage of patients received antipsychotics and what classes of antipsychotics. If the authors do not have such data, then the limitations section should be expanded.

Author Response

Comments 1: [I recommend correcting the visualization of ROC curves., ROC curves of indicators should be above the reference line according to generally accepted data representations. It is not difficult, only they need to change the data in the Value of State Variable line.]

Response 1: Thank you for pointing this out. We agree with this comment. Therefore, we made the change you mentioned. You can see this change in Figure 2a and Figure 2b.

Comments 2: [The authors indicate in the limitations section that therapy was not monitored. It is known that antipsychotic drugs significantly affect the concentrations of hormones of eating behavior, and metabolic syndrome is the main side effect of second-generation atypical antipsychotics. Given the long duration of schizophrenia in both groups, I would recommend separately presenting what percentage of patients received antipsychotics and what classes of antipsychotics. If the authors do not have such data, then the limitations section should be expanded.]

Response 2:  All patients in the acute phase included in the study had discontinued their medication for an unknown period of time and were not currently taking any medication. The medications used by patients in remission are shown in Table 1a.

However, in the limitations section, page 5, lines 150-157;

“In a study examining PYY in patients with schizophrenia; patients who received neuroleptics and those who did not were evaluated ; it has been stated that using neuroleptics has no effect on PYY values[14]. Although we argue that the deterioration in asprosin and PYY regulation is due to structural reasons in our study, the fact that patients are taking antipsychotics may still be a confusing factor. Although patients in the acute phase had discontinued the drug at the time of recruitment, it remains unclear how long they had been off the drug. Conducting this study in drug-free patients or designing the study in such a way that drug-related factors can be controlled may provide clearer interpretations.” as mentioned in the study.

The corrections suggested by all reviewers are written in red in the attached file.Please see the attachment.
Your evaluations are very important to us, thank you for your contribution.
Best regards...

Reviewer 2 Report

Comments and Suggestions for Authors
  • Introduction: The claim that asprosin and PYY are key regulators of energy metabolism in schizophrenia needs stronger support from prior literature.
  • Methods:
    • Mention the registration number of the ethical approval.
    • The manuscript states that written informed consent was obtained. However, was consent obtained from schizophrenia patients in the acute phase? If so, how was their capacity to consent assessed?
    • it is unclear whether statistical power calculations were performed for the primary outcome measures (asprosin and PYY levels). Please clarify further and explain explicitly.
    • There a lot of potential confounders that can affects metabolic parameters (potentially PYY and asprosin) e.g. antipsychotics, dietary intake, physycail activities, metabolic diseases (diabetes, obesity). The manuscript does not clearly specify clearly the patient condition and how these was accounted for in the analysis.
  • Results:
    • the AUC for PYY is only 0.607 (not statistically significant). This contradicts the claim that PYY can serve as a diagnostic marker. Be more cautious in the interpretation.
    • The study claims a "93% increase in PANSS scores per 1-unit decrease in asprosin." However, the β coefficient and confidence intervals should be presented.
    • The ROC curves (Figure 2a, 2b) should include confidence intervals.
    • The cutoff values for asprosin (7.39 ng/mL) and PYY (161.59 pg/mL) should be validated in independent cohorts before being proposed as diagnostic thresholds.
  • Discussion
    • PYY is described as "anorexigenic," yet the manuscript does not explore why it would be lower in schizophrenia, a disorder often associated with obesity and metabolic dysfunction. Are PYY receptors downregulated in schizophrenia? Provide more discussion.
    • The discussion section is still speculative, particularly in linking findings to neurodevelopmental hypotheses without direct evidence. A more balanced approach, acknowledging alternative explanations, would strengthen the manuscript.
    • Limittaions of the cross-sectional study design should be stated explicitly in the discussion.
    • If the authors aim to establish asprosin as a schizophrenia biomarker, a longitudinal study design (rather than cross-sectional) should be proposed for future studies to provide stronger evidence. Further, replication in drug-naïve schizophrenia patients would be necessary to rule out medication effects.
  • The conclusion should emphasize the need for future studies to validate these findings, address potential confounders, and explore underlying mechanisms.
Comments on the Quality of English Language

- Numerous grammatical and typographical errors reduce readability. A professional language edit is strongly recommended.

- The citation style is inconsistent, with missing DOIs for some references.

Author Response

Comments 1: [Introduction: The claim that asprosin and PYY are key regulators of energy metabolism in schizophrenia needs stronger support from prior literature.]

 Response 1: Thank you for pointing this out. We agree with this comment.The introduction has been expanded upon your suggestion by adding the following references. You can see it on Page 2 , lines 49-61.

Comments 2: [Mention the registration number of the ethical approval.]

Response 2: Thank you for pointing this out. We agree with this comment. The registration number of the ethical approval is 22943 on page 4, line 125, in the method section.

Comments 3: [The manuscript states that written informed consent was obtained. However, was consent obtained from schizophrenia patients in the acute phase? If so, how was their capacity to consent assessed?]

Response 3: Thank you for pointing this out. We agree with this comment. In the method section, page 4, lines 125-128, "Written informed consent was obtained from all participants and/or their court-appointed guardians, if any, and from the guardians of patients in the acute phase, stating that they agreed to participate in the study and gave permission to publish all clinical and other data in the article."; lines 143-144 "...….. their guardian agreed to participate in the study, and the guardian of patients in the acute phase agreed to participate in the study." the subject was tried to be clarified.

Comments 4: [It is unclear whether statistical power calculations were performed for the primary outcome measures (asprosin and PYY levels). Please clarify further and explain explicitly.]

 Response 4: Thank you for pointing this out. As per your suggestion; in the method section, page 4, line 130-136 "In the studies on asprosin and PYY in attention deficit hyperactivity disorder and schizophrenia patients, similar studies in the literature were taken as a basis and G*Power statistical program was used for sample calculation [10,15]. According to the sample calculations, it was determined that for a difference of 10 units to be significant, it would be appropriate to include at least 47 participants from each group at 95% power and 95% confidence interval (effect size = 0.70). Considering the possible loss of participants, it was planned to take 50 participants from each group." the subject was tried to be clarified.

Comments 5: [There a lot of potential confounders that can affects metabolic parameters (potentially PYY and asprosin) e.g. antipsychotics, dietary intake, physycail activities, metabolic diseases (diabetes, obesity). The manuscript does not clearly specify clearly the patient condition and how these was accounted for in the analysis.]

Response 5: Thank you for pointing this out. As per your suggestion; the following edits have been made to clarify the issue:

Methods:

“The inclusion criteria for all participants were to be between 18-65 years of age, have a BMI below 30, not have a diagnosed physical acute or chronic illness, not have an active alcohol-substance use disorder, not be pregnant or postpartum, agree to participate in the study and/or if they had a guardian, their guardian agreed to participate in the study, and the guardian of patients in the acute phase agreed to participate in the study.” (Page 4 line 141-144)

Discussions:

“In our study, no evaluation was made regarding the patients' nutritional styles. When evaluating appetite-related hormones; studies conducted with groups subjected to standard diets may provide more accurate data. Physical activity is known to affect metabolic parameters[32]. However, physical activity levels of the patients were not evaluated in our study. This may be a confounding factor to evaluate the results. Participants with a known physical disease and participants with a BMI above 30 were excluded from our study. However, possible unknown, undiagnosed metabolic diseases may have been missed during routine examinations.” (Page 5-6 line 159-167)

A new reference has been added.

  1. Lakka, T.A.; Laaksonen, D.E. Physical activity in prevention and treatment of the metabolic syndrome. Applied physiology, nutrition, and metabolism 2007, 32, 76-88. https://doi.org/10.1139/h06-113.

Comments 6: [The AUC for PYY is only 0.607 (not statistically significant). This contradicts the claim that PYY can serve as a diagnostic marker. Be more cautious in the interpretation.]

Response 6: Thank you for your consideration. Here, the AUC of 0.607 for PYY is the result of the analysis used to differentiate the acute phase among patients. According to this result, it was written that PYY is meaningless in analyzing the acute phase.

However, in the analysis with patients and controls, since the AUC for PYY was 0.962 and p<0.001, it was commented that it could identify the disease. 

Since the figures and tables explaining the data related to the figures were not consecutive in the text, we thought that they may not have been understandable.

We tried to arrange the figures and tables showing the ROC analysis consecutively.

Comments 7: [The study claims a "93% increase in PANSS scores per 1-unit decrease in asprosin." However, the β coefficient and confidence intervals should be presented.]

Response 7: Thank you for pointing this out. In Table 3, the β coefficient is given as -0.930.  Confidence intervals are added to Table 3.

Comments 8: [The ROC curves (Figure 2a, 2b) should include confidence intervals.]

Response 8: Thank you for your evaluation. As per your suggestion, confidence intervals are given in Figures 2a and 2b under ROC curves.

Comments 9: [The cutoff values for asprosin (7.39 ng/mL) and PYY (161.59 pg/mL) should be validated in independent cohorts before being proposed as diagnostic thresholds.]

Response 9: Thank you for pointing this out. We agree with this comment. Since our study is not a cohort study and is a cross-sectional study, the phrases 'can be used to identify' and 'can be used to distunguish disease' in the results have been replaced with 'associated with...' as much as possible, as follows:

“The results of the analysis to differentiate the acute phase show that PYY cannot be used to define the acute phase, but asprosine is associated with the acute phase in schizophrenia (Figure 2a). (Page 8, line 268)”

“The results of the analysis to differentiate disease showed that both PYY and asprosin were associated with disease (Figure 2b). (Page 8, line 272)”

In conclusion section;

“To confirm the findings of the study, more comprehensive independent cohort studies are needed in the future to elucidate the underlying mechanisms, especially by addressing possible confounding factors and assessing the limitations of the study.” has been added.

(Page 6, line 189-192)

Comments 10: [PYY is described as "anorexigenic," yet the manuscript does not explore why it would be lower in schizophrenia, a disorder often associated with obesity and metabolic dysfunction. Are PYY receptors downregulated in schizophrenia? Provide more discussion.]

Response 10: Thank you for your consideration. As per your suggestion, the following section has been added to the discussion section:

“Although obesity and metabolic dysfunction are common in schizophrenia, why both asprosin, an orexigenic peptide, and PYY, an anorexigenic peptide, are low should be addressed as a separate research topic. Studies on the subject at the receptor level may be useful.” (Page 6 , line 170-173)

Comments 11: [The discussion section is still speculative, particularly in linking findings to neurodevelopmental hypotheses without direct evidence. A more balanced approach, acknowledging alternative explanations, would strengthen the manuscript.]

 Response 11: Thank you for pointing this out. As per your suggestion, the sentences on the subject were tried to be constructed with adverbs 'probably' and modals 'may, may be', avoiding assertion as much as possible.

In addition, in the discussion part; the following sentences are used to balance the article:

“…In our study, asprosin levels were found to be associated with the disease and were also found to be different enough to be an indicator of the acute phase. However, the data obtained are not at a level that can be presented as a direct evidence.”(Page 4, line 85-88)

“Supporting the findings, which we think may be the result of a structural disorder, with neuroimaging studies can place the study on stronger foundations.”(Page 5, line 158-159)

Comments 12: [Limittaions of the cross-sectional study design should be stated explicitly in the discussion. If the authors aim to establish asprosin as a schizophrenia biomarker, a longitudinal study design (rather than cross-sectional) should be proposed for future studies to provide stronger evidence. Further, replication in drug-naïve schizophrenia patients would be necessary to rule out medication effects.]

 Response 12: Thank you for pointing this out. Upon your suggestion, the following part has been added to the discussion section:

“The cross-sectional design of the study is our main limitation. The cross-sectional design limits the ability to infer causality between asprosin and PYY levels and disease. The cross-sectional design does not provide data on the temporal dynamics of these biomarkers. Conducting a study in patients with first attack schizophrenia in order to see the disease-defining effect of asprosin and PYY will give more reliable results. Again, designing a study in which the values of the same patient in both the acute phase and the remission phase can be compared can provide a more accurate interpretation of the data. In a study examining PYY in patients with schizophrenia; patients who received neuroleptics and those who did not were evaluated ; it has been stated that using neuroleptics has no effect on PYY values[14]. Although we argue that the deterioration in asprosin and PYY regulation is due to structural reasons in our study, the fact that patients are taking antipsychotics may still be a confusing factor. Although patients in the acute phase had discontinued the drug at the time of recruitment, it remains unclear how long they had been off the drug. Conducting this study in drug-free patients or designing the study in such a way that drug-related factors can be controlled may provide clearer interpretations.” (Page 5, line 143-157)

Comments 13: [The conclusion should emphasize the need for future studies to validate these findings, address potential confounders, and explore underlying mechanisms.]

Response 13: Thank you for pointing this out. Upon your suggestion, the following part has been added to the conclusion:

“To confirm the findings of the study, more comprehensive independent cohort studies are needed in the future to elucidate the underlying mechanisms, especially by addressing possible.”(Page 6, line 189-192)

Comments 14: [Numerous grammatical and typographical errors reduce readability. A professional language edit is strongly recommended.]

Response 14: Your evaluation is very valuable for us, thank you. We have tried to correct grammatical and typographical errors.

Comments 15: [The citation style is inconsistent, with missing DOIs for some references.]

Response 15: Your evaluation is very valuable for us, thank you. The references have been reviewed again in line with your suggestion. Only references 11 and 19 could not be added because they do not have DOIs.

The corrections suggested by all reviewers are written in red in the attached file.Please see the attachment.
Your evaluations are very important to us, thank you for your contribution.
Best regards...

Round 2

Reviewer 2 Report

Comments and Suggestions for Authors

The authors have addressed my remarks sufficiently.

Author Response

Thank you.
Best regards...